

# The effect of farmland on the surface water of the Aral Sea Region using Multi-source Satellite Data

Jiancong Shi[1,2], Qiaozhen Guo[1], Shuang Zhao[1], Yiting Su[3] and Yanqing Shi[1]

[1] School of Geology and Geomatis, Tianjin Chengjian University, Tianjin, China
[2] Coal Industry Taiyuan Design and Research Institute Group Co., Ltd., Taiyuan, China
[3] Department of Surveying and Land Use, College of Geoscience and Surveying Engineering, China University of Mining and Technology (Beijing), Beijing, China

## ABSTRACT

**Background:** The improper land utilization has brought tremendous pressure on the surface water of the Aral Sea Region in the past decades. It was seriously hindered for construction of the Green Silk Road Economic Belt by the fragile environment. Therefore, it is of great necessity for environmental protection and social development to monitor the change of surface water in the Aral Sea Region.

**Methods:** In this study, LandTrendr algorithm was used on Landsat time-series data to characterize the change in farmland on the Google Earth Engine platform. Based on multi-source data, the water area changes of the Aral Sea were extracted based on the Google Earth Engine, and the mean method was utilized to extract the changes in water level and water storage. Finally, a water-farmland coupling degree model was utilized to evaluate the impact of farmland changes on the surface water in the Aral Sea Region.

**Results:** As a result, the change of farmland is as follows: the farmland area of the Aral Sea Region has abandoned 3,129 km$^2$ from 1987 to 2019, with overall accuracy of 85.3%. The farmland change had increased the drainage downstream of the Amu Darya River and the Syr Darya River. It has led area of the Aral Sea to decrease each year continuously. The area of the Aral Sea shrank by 1,606.36 km$^2$ per year from 1987 to 2019. Furthermore, Aral Sea's water level decreased by 0.13 m per year from 2003 to 2009. The amount of water storage in the Aral Sea Region also showed a downward trend from 2002 to 2016. There was a high-quality coupling coordination 0.903 relationship between surface water and farmland. It will increase the burden of water for people's normal daily life by the water loss resources caused by abandoned farmland. This study emphasized threat of unreasonable farmland management to surface water of the Aral Sea Region. The findings contributed for decision makers to formulating effective reasonable policies to protect surface water and use land of the Aral Sea Region. Meanwhile, the application of coupling degree model can provide a new method for studying the connection of independent systems in the farmland, water, environment and more.

Corresponding author
Qiaozhen Guo,
gqiaozhen@tcu.edu.cn

## INTRODUCTION

As the essential water resources in the Aral Sea Basin and even in Central Asia, the Aral Sea could primarily ensure the stability of the surface water to be maintained (*Micklin, 1988*). However, the area of the Aral Sea is currently only one-tenth of that in 1960 (*Wu et al., 2020*). As impacted by the rapid shrinking of the Aral Sea, the surface water of the Aral Sea Basin has been severely decreased (*Eleni et al., 2020*). The shrinking surface water caused land desertification and land salinization (*Parajuli & Yang, 2017*), thereby seriously endangering the development of society. Accordingly, it is essential for protection of the surface water and regional development to reveal the root cause of the Aral Sea's shrinkage.

The Aral Sea Basin refers to a severely arid area with annual precipitation of less than 300 mm (*Elena, 2014*). Thus, the level of the Aral Sea is determined by the inflow of two feeding rivers, the Amu Darya and the Syr Darya (*Philippe et al., 2007*). The two rivers originate from the melting of snow in the mountains of southeast of the Aral Sea Basin (*Micklin, 2016*). Scholars have carried out researches on the upstream of Amu Darya and Syr Darya, as presented below. Valentina et al. found that the snow depth in the rivers' (*e.g.*, the Amu Darya and the Syr Darya) source tended to decrease after 1965, whereas the runoff downstream has been declining (*Valentina & Ladislav, 2009*). *Hagg et al. (2007)* considered that the runoff of the upper reaches in the Amu Darya River would gradually increase. In a word, considerable researches indicated a downtrend of snow depth in the source of the Amu Darya River and the Syr Darya River, whereas the runoff downstream has been declining (*Annina et al., 2012*; *Wang et al., 2016*). The above studies show that the upstream runoff of Amu Darya and Syr Darya increases each year, which is contrary to the shrinkage of the Aral Sea. At the same time, researchers have found that the shrinkage of the Aral Sea is directly related to the reduction of runoff downstream of rivers (*Shi et al., 2021*; *Micklin, 2007*). Therefore, this study will focus on the downstream area of rivers near the Aral Sea.

Over the past decades, unreasonable land utilization led to the decline of downstream runoff in the Amu Darya River and the Syr Darya River, which directly caused the shrinking of the Aral Sea (*Christopher et al., 2016*). Primarily, neighbouring countries used a lot of lands to plant farmland, and have done many radical measures to increase the value of agricultural output. For instance, the former Soviet Union built the channel with 500 km in the Amu Darya River area and the Syr Darya River area in 1960. Such a project took one-third of the water in rivers downstream to farmland (*Conant, 2006*). The mentioned measure altered the rain-fed irrigation mode of farmland in the Aral Sea Basin (*Micklin, 2007*). By 2010, the irrigated area of the Aral Sea Basin was 7,895,600 ha, taking up 78.6% of planted land (data from the Central Asia Water Resources Network: http://cawater-info.net/). Inefficient irrigation systems and terrible management

of land utilization triggered the increase in water withdrawn from rivers. Thus, surface water protection requires the exploration of the farmland variations in the Aral Sea Basin. Several studies have been conducted on the surface water variations attributed to irrigation. For instance, *Jin et al. (2017)* investigated human irrigation-induced hydrological variations in the Aral Sea Basin by multiple satellite data. *Fabian et al. (2015)* mapped abandoned farmland to support surface water protection in Kyzyl-Orda, Kazakhstan. However, existing studies could not determine a quantitative model to assess the impact of farmland changes on surface water. Accordingly, it is of great necessity to establish a method to assess the effect of farmland changes on surface water.

For the complicated political issues and poor physical accessibilities of the Aral Sea Region, there are severe limitations in collecting data required for large-scale and long-term change detection and analyses (*Fabian et al., 2015*). As an alternative, remote sensing technology, exhibiting the advantages in data of effective coverage of long-term span and multi-time scale, has been extensively employed in natural disaster early warning, eco-environmental protection, farmland protection, dynamic analysis of land use, social development, etc. (*Tewkesbury et al., 2015*; *Li & Narayanan, 2003*; *Saha, Bovolo & Bruzzone, 2019*). *Singh, Seitz & Schwatke (2012)* used multi-source data to explore inter-annual water storage variations in the Aral Sea. *Jin et al. (2017)* adopted multiple satellite data to view environmental variations around the Aral Sea. At present, remote sensing technology has become an essential measure of long-term monitoring variations in surface water in large areas (*Singh, Seitz & Schwatke, 2012*). However, the above researches are difficult to characterize the impact of farmland on the surface water quantitatively. Thus, this study combined the comprehensive variations of water and farmland changes to explore the internal relationship between the surface water and farmland based on multi-source data.

The water area, water level and surrounding water storage of the Aral Sea act as critical indicators to characterize the hydrological variations in the Aral Sea. Multi-source data efficiently supports extracting the comprehensive variations of water. The LandTrendr (Landsat-based Detection of Trends in Disturbance and Recovery) algorithm can capture both long-term gradual and short-term drastic variations (*Gorelick et al., 2017*). Moreover, *Kennedy et al. (2018)* encapsulated the LandTrendr on the GEE platform to consume considerable computation time in continuous change detection. Moreover, it has been broadly applied in most fields of change detection (*Zhu et al., 2019*). The coupling degree model is capable of reflecting the degree of association between multiple subsystems.

This study aimed to examine the surface water attributed to land utilization change of the Aral Sea Region. The specific steps are as follows: Firstly, the LandTrendr algorithm was run on Landsat time-series of the GEE platform to detect the land utilization change modes. Second, the variations of water area, water level and water storage were detected using multi-source data (Landsat, ICESat, GRACE). Lastly, to explore the effect of farmland changes on the surface water, the variations of water (water area, water level and water storage) and farmland were combined to build a water-farmland coupling degree model. The mentioned efforts can provide explicit cognitions regarding the effect of

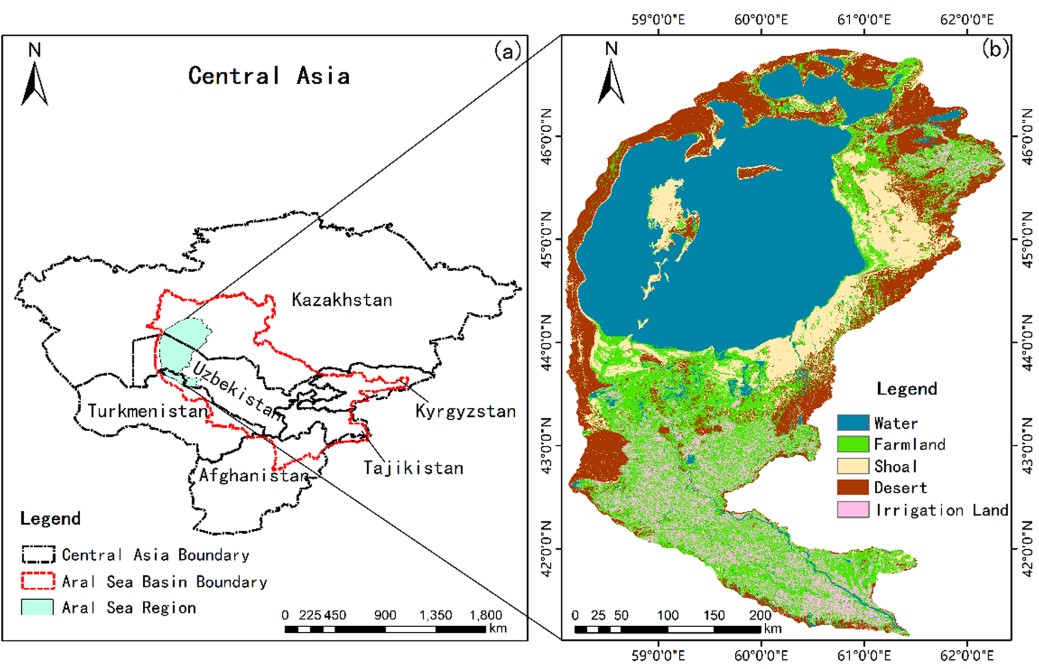

**Figure 1 Research scope.** (A) General location in the Aral Sea Basin and the Central Asia; (B) 1987 land-use classification from Google Earth Engine (GEE) imagery .

farmland on surface water. Furthermore, it can provide practical implications for agricultural governance, water protection and surface water balance.

# MATERIALS AND METHODS

## Study area

The Aral Sea Region is located on the border between northwest Uzbekistan and southwest Kazakhstan. It covers 130,000 km$^2$, and includes North Aral, East Aral, West Aral and irrigated farmland of the Aral Sea Region (Fig. 1). This is a typical arid area with an average annual temperature of 11.62 °C and precipitation of 126.26 mm.

## Data preparation

### Landsat data and pre-processing

Landsat surface reflectance data (Landsat5, Landsat7 and Landsat8) were collected from 1984 to 2019 on GEE platform. This study used the FMASK and neighboring position filling to improve extraction accuracy to remove the mentioned poor-quality (snow/ice, clouds, building shadows and scan-line corrector (SLC)-off gaps) images (*Zhu & Woodcock, 2012*). Accordingly, the Landsat surface reflectance dataset was produced for our study area from 1987 to 2019 (A-LSTC). Lastly, the dataset (YA-LSTC) was synthesized into annual images from the A-LSTC (June 1 to September 1). The normalized vegetation index (NDVI) was introduced to the dataset as a new band (*Tucker, 1979*).

$$NDVI = \frac{\rho_{nir} - \rho_r}{\rho_{nir} + \rho_r} \qquad (1)$$

where $\rho_{nir}$ and $\rho_r$ are the near-infrared band and red band in Landsat.

### ICESat satellite altimetry data

The ICESat mission was initiated in January 2003 and ended in February 2010 by National Aeronautics and Space Administration (NASA) (*Schutz et al., 2005*). The Geoscience Laser Altimeter System (GLAS) on ICESat has provided global measurements of polar ice sheet mass balance, cloud and aerosol heights, land topography and vegetation characteristics with a surface sampling diameter of 70 m as well as a spacing of 172 m (*Zwally et al., 2008*). ICESat/GLAS level 2 altimetry product (GLA14) consists of global land surface elevation data (*Kwok, Zwally & Yi, 2004*). The GLA14 has been broadly introduced in lake water level and land elevation and have reflected accuracy over 10 cm (*Zhang et al., 2011*). Thus, the GLA14 can precisely contribute to the surface elevation. The GLA14 from 2003 to 2009 were collected to extract the water level change in the Aral Sea.

### GRACE data

The Gravity Recovery and Climate Experiment (GRACE) mission was launched in March 2002 and ended in October 2017 by NASA. The water mass variations were considered a change of Earth's gravity by GRACE (*Jin et al., 2017*). Most gravity signals after de-noising of GRACE reflect variations in water storage (*Chen et al., 2019*; *Swenson & Wahr, 2007*). However, the orbital altitude and resolution of GRACE limited the effective research range (<200,000 km$^2$) (*Singh, Seitz & Schwatke, 2012*). Thus, the original research range was extended to 230,000 km$^2$. Though the study area is less than 230,000 km$^2$, it can be assumed that the mass variations on long-time scales come from the long-term storage change of water of the Aral Sea Region. Thus, the GRACE data was processed in the GEE platform from 2002 to 2016 (research range between June and September each year).

## Detection algorithm
### Detection algorithm of farmland change

LandTrendr (Landsat-based Detection of Trends in Disturbance and Recovery) algorithm is an approach to extract the spectral trajectories of land surface from Landsat time-series stacks (*Watts & Laffan, 2014*). This algorithm can reflect farmland variations by detecting the temporal-spectral trajectory of each pixel. The input of every pixel for the algorithm is a time series with one spectral value or index. This algorithm also needs to set parameters, such as noise-induced spikes (outliers), potential vertices (breakpoints), fitting trajectories, and the optimal number of segments (*Watts & Laffan, 2014*).

LandTrendr algorithm on the GEE platform was used to detect the farmland change in the Aral Sea Region (Fig. 2A). Before running this algorithm, the features of farmland variations were analyzed, tested different combinations of parameter values. The NDVI was determined as the spectral index, and other parameters were calculated (Table S1).
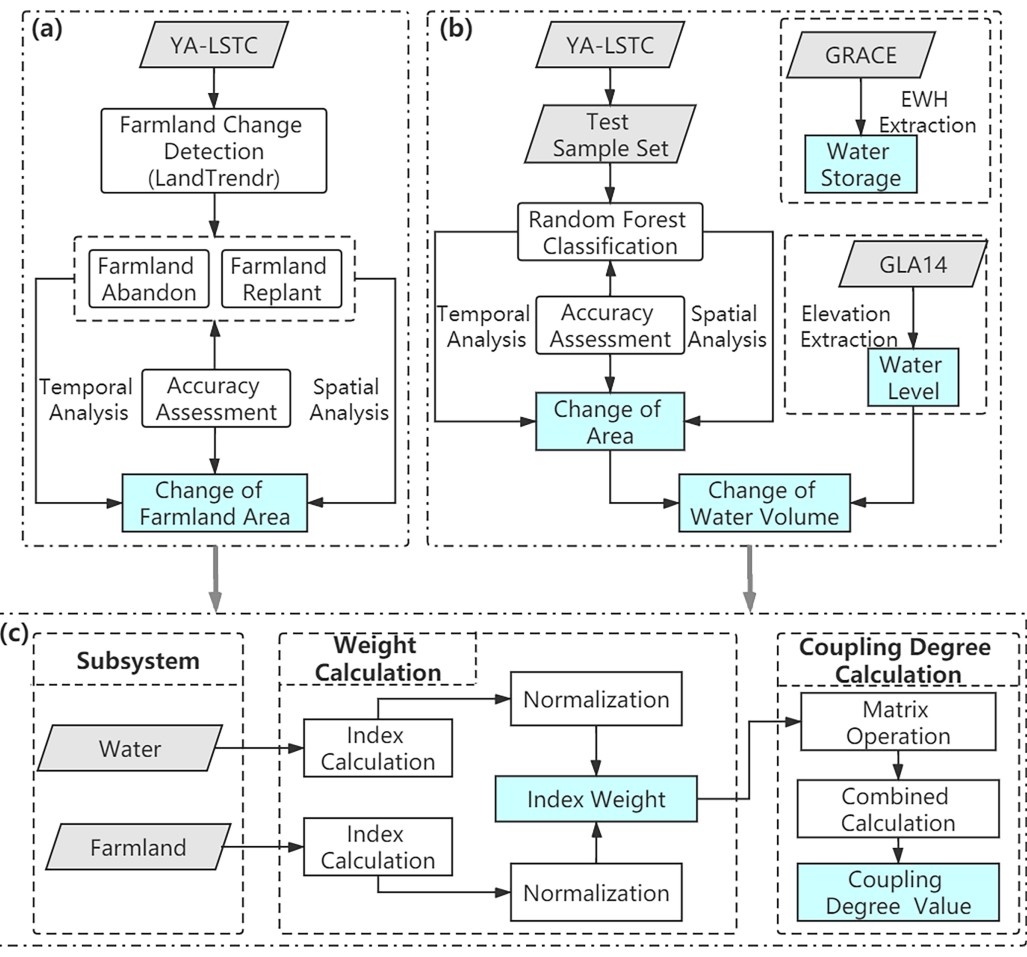

**Figure 2 Research method.** (A) The method of farmland change detection; (B) the method of water change detection; (C) the method coupling degree model.

NDVI of farmland ranged from 0.45 to 0.69 in the Aral Sea Region and plummeted after abandonment to range from 0.2 to 0.3. Thus, NDVI of abandoned farmland was set between 0.2 and 0.3. The mentioned thresholds were adopted to detect the change patterns of farmland, the change time of occurrence and end, and duration.

### Detection algorithm of waters change

*Via* the GEE platform, the random forest algorithm was adopted to detect the variation of water area (Fig. 2B). Random forest is a machine learning algorithm that is complies with the decision tree. This algorithm uses CART clustering for prediction and classification (*Breiman, 2001*). Two primary parameters should be set before implementing the random forest algorithm. To be specific, the first refers to the number of splitting times (mtry) of the branch tree in each model. This parameter should be set as the square root value of variables and 2 times or 1/2 of the prediction's open value. In this paper, the variables specifically refer to the bands of remote sensing images (Table S2). The second refers to the number of generated trees (ntree) in model operation, and the amount of model calculation is proportional to the value of ntree. The ntree was set as 500 because it will be

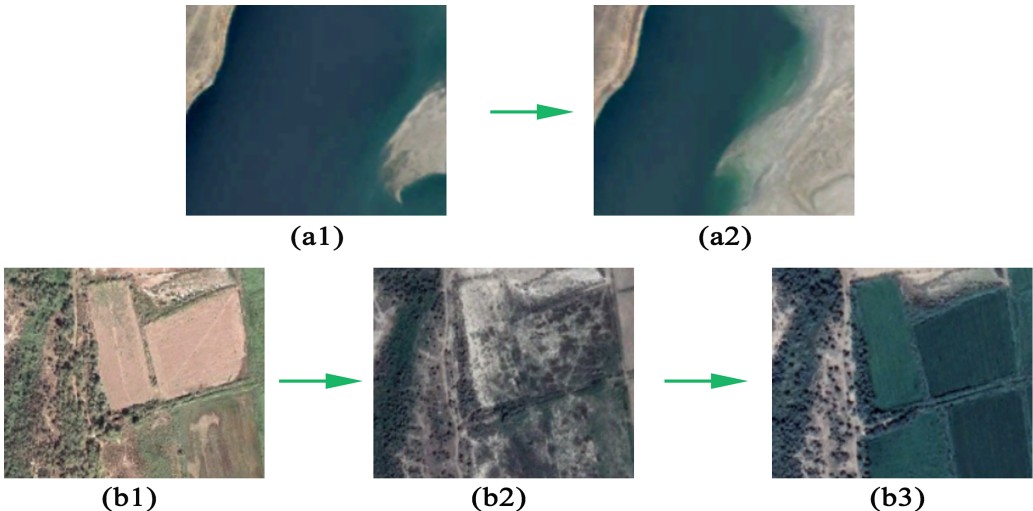

**Figure 3 Examples areas of water change and farmland change near the Syr Darya and the Amu Darya.** In the figure, a1 and a2 represent the water change years 2009 and 2019; b1, b2, b3 represent the water change years 2009, 2016 and 2019.

pretty obvious with a bit of tinkering that the predictions from the model will not change much after this value.

The accuracy of model depends more on quality of training samples. Therefore, we selected 200 pixels representing water and another 200 pixels representing non-water based on Landsat imagery. In order to access accuracy of results, we visually examined all selected pixels (from 1987 to 2019) and used high-spatial-resolution Google Earth imagery for further manual interpretation of land cover (Fig. 3). Therefore, we determined and recorded whether and when water pixels change occurred.

The ICESat data was adopted to analyze the alteration in water level in the Aral Sea. At first, the data format of GLA14 was converted. Second, GLA14 was adopted to extract the water level of the Aral Sea (Eq. (2)). Lastly, the water volume was calculated according to the alteration in water level and the variation of water area of the Aral Sea (Eq. (3)). Meanwhile, researchers compared ICESat data with airborne LiDAR data and DEM from GPS data. The error of ICESat data is less than 3 cm (*Kurtz et al., 2008*; *Fricker, 2005*).

$$WL_T = \frac{\sum_{i=1}^{n} wl}{n} \tag{2}$$

$$WV = WL_T * WA \tag{3}$$

where $WL_T$ is mean water level. $wl$ is the land elevation of every ICESat point. $WV$ is water volume. $WA$ is water area.

The GRACE dataset was used to extract the equivalent water height (EWH). The data contained in this dataset are units of "Equivalent Water Thickness" which represent the deviations of mass in terms of the vertical extent of water in centimeters (*Watkins et al., 2015*). The EWH was processed at NASA Jet Propulsion Laboratory. Actually, this dataset was produced by three institutions: CSR (U. Texas/Center for Space Research),

GFZ (German Research Center for Geosciences), and JPL (NASA Jet Propulsion Laboratory). However, since each institution independently produces the coefficients, the results may be slightly different. And the mass change maps of the three institutions have typically suffered from poor observability of east-west gradients, resulting in "N-S stripes" that are conventionally removed *via* empirical smoothing and/or "destriping" algorithms. In order to overcome these limitations, Watkins et al., developed GRACE(-FO) mass concentration (mascon)solutions (*Watkins et al., 2015*; *Wiese, Landerer & Watkins, 2016*; *Landerer et al., 2020*). The solutions allow for convenient application of *a priori* information derived from near-global geophysical models to prevent striping. Significantly, over the land, mascon solutions have more excellent resolution for smaller spatial regions, in particular when studying secular signals. Meanwhile, for different application scenarios, the dataset has its advantages. For instance, it is more suitable for water, ice mass, and ocean bottom pressure mass change application to use JPL mascon (*Wiese, Landerer & Watkins, 2016*; *Landerer et al., 2020*). Therefore, JPL data, which have been processed by the GRACE(-FO) mascon solutions, were used in this paper. And the measurement accuracy of GRACE data is credible (5,000 km wavelength: 0.001 mm; 500 km wavelength: 0.01 mm) (*Zheng et al., 2009*). Water storage variation in the Aral Sea Region was analyzed (Eq. (4)).

$$WS = EWHT * A \tag{4}$$

where EWHT is the equivalent water height from GRACE dataset; WS is water storage; A is the area of study region.

## Accuracy assessment

Two hundred pixels were selected to build a confusion matrix (CM) to assess the accuracy of the five surface features in 1987 (*i.e.*, water, farmland, desert, shoal and irrigation land). Meanwhile, we visually examined all surface features by high-spatial-resolution Google Earth imagery. The overall accuracy, user accuracy, and producer accuracy for each feature were concluded following the indexes defined by Foody and Green (*Congalton & Green, 1993*; *Foody, 2002*). Two hundred validating pixels were selected to represent the stable area of farmland. In addition, another 200 validating pixels represent the various area of farmland for each change mode. The stable area refers to the pixels that remain unchanged as farmland for years. While the various area represents the changed farmland pixels during the whole study period. The 20% pixels of training samples in water extraction were adopted to produce a CM to access the accuracy of water extraction results.

## Water-farmland coupling degree model

The coupling degree model originated from the coupling coefficient model in physics. Farmland and surface water were combined to build the water-farmland coupling degree model to explore the impact of farmland on surface water. In such a model, four indicators were selected related to surface water and farmland (Fig. 2C). At first, every indicator was normalized, and then the entropy method was used to determine their weight. The index matrix is $X = (x_{ab})_{i \times j}$ (where i is the evaluation index; j is the index, and $x_{ab}$ is

the b-th index of the a-th sample,). The coupling degree model (C) is as follows (*Chang, 2018*).

$$C = \left\{ \frac{f_1(x) * \cdots * f_i(x)}{\left| \frac{|f_1(x) + \cdots + f_i(x)|^i}{i} \right|} \right\}^{1/i} \tag{6}$$

$$f_i(x) = \sum_b^j w_b x_b \tag{7}$$

where $f_i(x)$ is the comprehensive evaluation function of subsystem; $w_b$ is the weight of each index; $x_b$ is each index; $i$ is the number of subsystems to be access. There are two subsystems which need to be evaluated, so $i = 2$. ($C \in [0, 1]$). If the C value is closer to 1, it will mean that the subsystems are more related and coordinated (*Congalton & Green, 1993*).

# RESULTS

## Accuracy assessment

The CM for the classification of 1987 imagery (Table S3) revealed that the overall accuracy (OA) reached over 90%. The user accuracy (UA), and producer accuracy (PA) for each class mainly were more than 86%. The accuracy of farmland is less than 90%. The reasons are elucidated below. There were considerable irregular water bodies and farmland in irrigated land of the Aral Sea Region; thus, the farmland often tended to mix with water and Irrigation land. Moreover, some sparse vegetation was planted in the irrigation land. Thus, it was difficult to distinguish by employing Landsat images with a spatial resolution of 30 m. However, the overall accuracy reached 92.1%, suggesting the random forest model's capability. A similar method was used to assess the accuracy of extraction of water area in the Aral Sea from 1987 to 2019. The accuracy of extraction of water area reached over 90%, demonstrating the credibility of the results.

Table S4 show the farmland detection accuracy of abandon and replanting by LandTrendr. The overall accuracy of 85.3% was low for abandoned farmland for the mixture of abandoned farmland and sparse planting. For the replanting, the PA, UA, and OA are above 80%. Thus, it is sufficient for farmland to detect the change model. In summary, the OA of abandoned farmland change detection was more than 85%, demonstrating the detection method's reliability based on the LandTrendr.

## Temporal and spatial variations of farmland

The distribution of abandoned farmland, in the Aral Sea Region during 1987–2019, was rendered the disturbance map with a gradient (Fig. 4) according to the occurrent years. The abandoned region was mainly diffused around the Syr Darya and the Amu Darya, especially around the Amu Darya (Table S5). Within 9 km of these Draya rivers, the percentage of abandoned farmland reached 59.96% from 1987 to 2019. Overall, 3,129 km$^2$

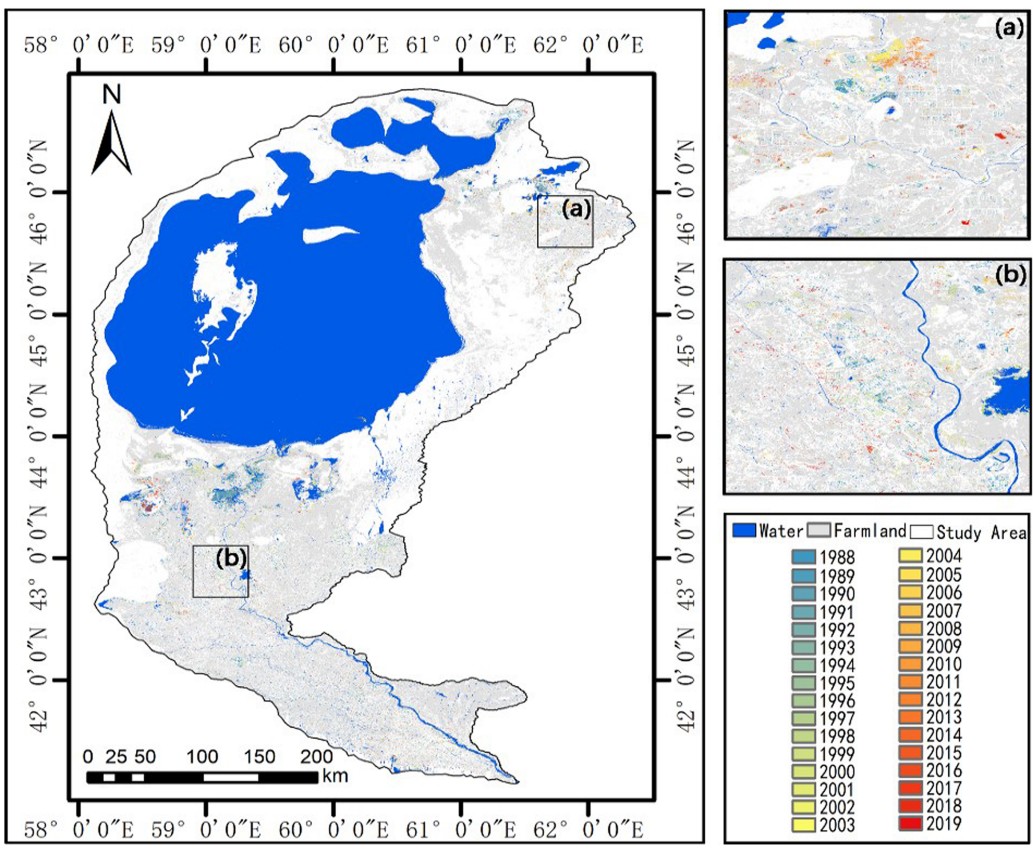

**Figure 4 Year of abandoned farmland in the Aral Sea Region with two areas shown in detail.**
(A) Around the Syr Darya River; (B) around the Amu Darya River.

of farmland was abandoned. The annual change peaked between 1987 and 1993 (Table S6), after which percentage of abandoned farmland decreased under 3% from 2002 to 2016 (Table S6). However, the percentage of abandoned farmland was elevated to 5.7% in 2017 (Table S6), which resulted in the plummet of water volume in the Aral Sea.

## Waters variations of the aral sea region

The Aral Sea has been shrinking at 1,606.36 km$^2$ per year from 1987–2019. In 2019, the area of the Aral Sea took up only 15.6% of its size in 1987 (Fig. 5). With an annual decrease of 1,306.18 km$^2$, the East Aral (820.22 km$^2$) had almost dried up by 2019. The area of the East Aral, in 2019, only accounted for 1.92% in 1987. On the contrary, as impacted by the construction of the Dike Kokaral dam in 2005, the North Aral showed a weak expansion trend with an average annual growth of 10.05 km$^2$. The area of North Aral rose to 4,658.773 km$^2$ in 2019. Kazakhstan Government built the Kokaral dam at the junction of the North Aral and East Aral (Fig. 5). In order to keep the water level of North Aral and environment stable, the water flow from North Aral to East Aral was artificially reduced. Though this policy effectively prevented the loss of North Aral, it also reduced the water flowing into East Aral. Such a dam aggravated the dry-up of East Aral. The West

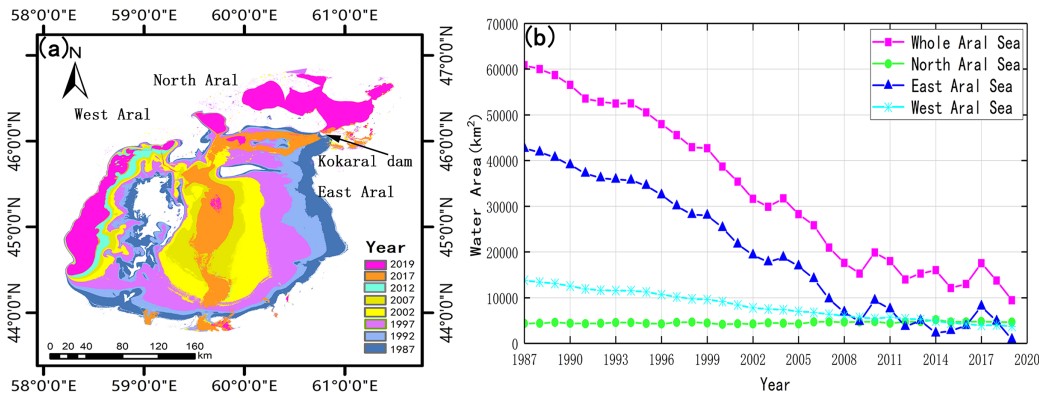

**Figure 5 Change of the Aral Sea area from 1987 to 2019.** (A) The spatial distribution; (B) the temporal distribution.

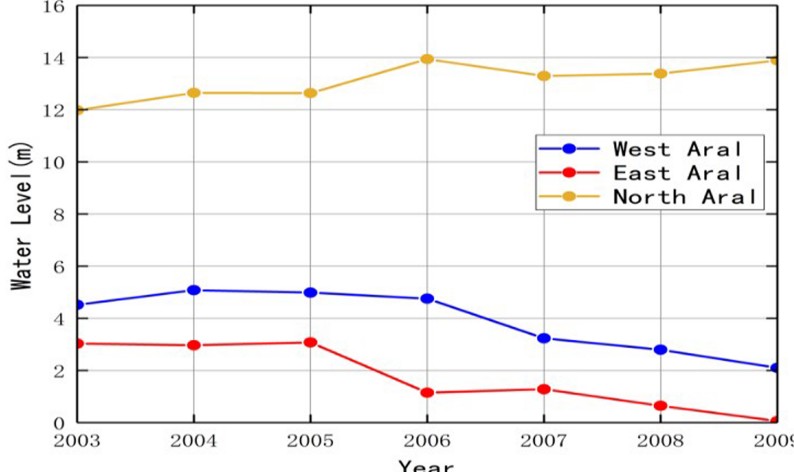

**Figure 6 Change of the Aral Sea water level from 2003 to 2009.**

Aral is small and far from the irrigated area. Accordingly, the West Aral had shrunk more minor than the East Aral at 314.83 km² each year.

Likewise, the water level of the Aral Sea tended to decrease. The curve of the water level of the North Aral significantly ascended as impacted by the construction of the dam (Fig. 6). The water level of the North Aral was elevated near 0.32 m per year from 2003 to 2009. In the East Aral, the water level drastically declined at 0.5 m per year. Moreover, the East Aral was only 0.05 m in 2019. Compared with the East Aral, declining West Aral was lower as impacted by its smaller range and farther from the irrigated area. To be specific, the water level of the West Aral decreased at 0.4 m per year and dropped to 2.1 m in 2019.

Figure 7 illustrates the water mass change of the GRACE analysis in the study area. The curve of equivalent water height shows a downward trend from 2002 to 2016. The change of equivalent water height is same as the water level. The water mass was overall declining, marking a decrease by 0.97 km³ each year. The decrease in water mass

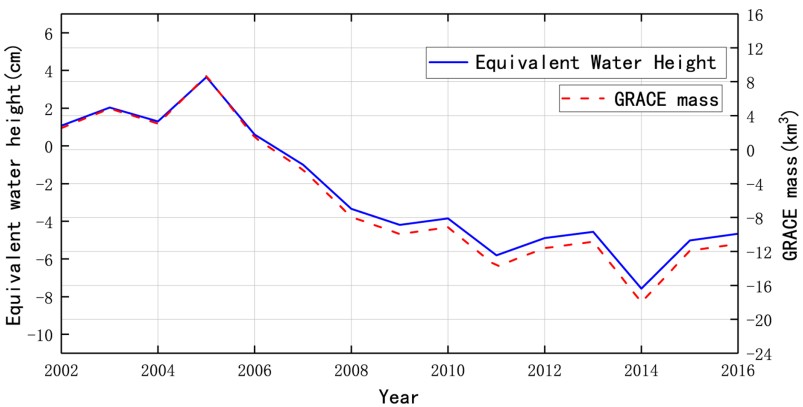

**Figure 7 Mass change of the Aral Sea Region from 2002 to 2016.**

represented the descending of water storage, such as the Aral Sea, rivers, reservoirs and groundwater.

## Construction of water-farmland coupling degree model

According to the results of this study regarding water and farmland in the Aral Sea Region, three indexes were selected to build a comprehensive evaluation index system (Table S7). This system included two subsystems (surface water system, farmland system). Water subsystem had two evaluation indexes, and farmland subsystem had one evaluation index (Table S7).

The entropy method was adopted to calculate the weight of each index. According to the mentioned weights, the coupling degree was obtained between the two subsystems from 2003 to 2009. In addition, the results indicated that the annual coupling degree value was above 70%, and the average coupling degree reached 0.903 (Table S8). By the criterion of coupling degree (Table S9) (*Chang, 2018*), only 2003 and 2005 pertain to intermediate level coupling coordination. Other years belong to high-quality coupling coordination. In addition, the average coupling degree belongs to the high-quality coupling coordination. The mentioned coupling degree results suggested that the greater coupling degree between the two subsystems, and farmland changes affected the variations of surface water in the Aral Sea Region.

## DISCUSSION

### Benefits of methods

The results shown that, the loss of water by abandoned farmland will increase the burden of normal water use. During the past years, many scholars were committed to analyzing the reasons for the decrease of surface water in the Aral Sea Region. Compared with previous researches, our work has similarities and differences. For similarities, the surface water loss was caused by improper irrigation management. On the contrary, our research was from the perspective of farmland abandonment. Few studies have analyzed the loss of water caused by abandoned farmland. They were analyzed effect of

land water storage on surface water (*Jin et al., 2017*; *Gaybullaev, Chen & Gaybullaev, 2012*; *Parajuli & Yang, 2017*).

The coupling degree model was used for the first time to monitor the effect of farmland on the surface water of the Aral Sea Region. Most previous researches regarded farmland and surface water as independent objects, which could not acquire the natural connection between them. Therefore, changes in farmland and surface water were put into a quantifiable system that can be used to explore the substantial impact of farmland changes on the surface water. According to the result, the coupling degree model showed good applicability in this field. In addition, it could provide a new method for the internal connection between farmland, surface water, and ecology in the Aral Sea Region and even the Aral Sea Basin.

## Effect of abandoned farmland on surface water in aral sea region

The abandoned farmland will cause a lot of surface water loss. The main reason is that irrigation canals of abandoned farmland are not blocked because of the negligence of management. It directly led to majority of the water was being soaked up by the desert and blatantly loss (*Ibragimov et al., 2007*; *Maksud et al., 2016*; *Jabbarov et al., 2013*). According to the results of section 3.2, we extracted the area of abandoned farmland (Table S6). The area of abandoned farmland reached per year 97.843 km$^2$. And the abandoned farmland will cause a lot of water waste. Farmland in Aral Sea Region is concentrated on planting cotton, wheat and rice (*Chapagain, Gautam & Hoekstra, 2005*). Table S10 shows that wheat is the thirstiest crop. And the water use efficiency of cotton is the lowest (0.451 * 108 kg/km$^3$). Table S11 shows the crop production in precious decades.

Based on the above results, the loss volume of water loss by abandoned farmland can be extracted. If all the abandoned farmland were used to plant cotton in the past few years, the water loss would reach 0.028 km$^3$ every year (Table S12). According to the current distribution of available water *per capita* from the World Bank (Uzbekistan, 2,596 m$^3$/per person/annually; Kazakhstan, 1,943 m$^3$/per person/annually), the water loss can provide annual domestic water for tens of thousands of people in the Aral Sea Region. The specific numbers are as follows: Kazakhstan are 14,410 persons, and Uzbekistan are 10,785 persons. Therefore, abandoned farmland caused additional loss of normal domestic water for people in the Aral Sea Region.

The previous researches gave us much inspiration. Micklin macroscopically analyzed the reason and effect of water changes in the Aral Sea (*Micklin, 2010*). *Shibuo, Jarsjö & Destouni (2007)* found nearly 100% return flows of the applied irrigation water in the Aral Sea Drainage Basin. *Cretaux, Letolle & Bergé-Nguyen (2013)* analyzed effect of groundwater on the hydrological water budget of the Aral Sea by remote sensing data. Compared with above researches, our contributions are as follows:

1. Based on the coupling degree model, we quantitatively characterized the effect of farmland changes on surface water.
2. The coupling degree model was used for the first time to study the effect of farmland changes on surface water. We believe that this attempt will help broaden the thinking of

researchers in the fields of remote sensing, environment and more. They can connect independent systems through the coupling degree model.

3. The effect mechanism was analyzed on surface water changes by calculating the amount of water loss in abandoned farmland. Moreover, the normal domestic water for people caused by water loss was analyzed.

### Limitations of research

At first, vegetation indices can reduce other factors' impacts in monitoring land cover change (*Zhang et al., 2013*). The NDVI can reduce the influence of external environmental factors on the surface reflectance to some extent. Previous studies indicated that the accuracy of NDVI in detecting farmland is better than others (*Tong et al., 2017*). However, due to the disturbance by other vegetation in the arid region, using NDVI to the LandTrendr algorithm has some limitations (*Watts & Laffan, 2014*). A typical example is that the sparse vegetation of the Aral Sea Region generated the same NDVI signal with farmland and affected the detection results. Therefore, it is necessary to add additional indices to improve accuracy.

Secondly, the data on water level and water storage cannot match the water area's time series. However, this fusion of data yet offers valuable and unique information on the surface water. Simultaneously, in the next step of our research, we need to improve data integrity of the surface water.

## CONCLUSIONS

In this study, the farmland variations were detected by using LandTrendr algorithm based on the GEE platform. Subsequently, multi-source data (Landsat, ICESat, GRACE) were employed to extract the variations of water area, water level, water storage in the Aral Sea Region. Given the mentioned results, the Water-Farmland assessment was developed. Such a system assessed the impact of farmland on surface water in the Aral Sea Region. The specific results are elucidated below:

Throughout the research, the abandoned farmland areas covered 98 $km^2$ per year. The overall accuracy of abandon took up 85.3%. There was most abandoned farmland near the rivers (the Amu Darya River and the Syr Darya River) from 1987 to 2019 of the Aral Sea Region ("Temporal and Spatial Variations of Farmland"). The abandon of farmland had triggered a massive loss of water resources of the Aral Sea Region.

Second, the Aral Sea shrank by 1,606.36 $km^2$ per year from 1987 to 2019, in which the East Aral shrank most seriously. Furthermore, the Aral Sea's water level was down-regulated by 0.13 m per year from 2003 to 2009, especially the East Aral. However, as impacted by the construction, Dike Kokaral dam in 2005, the water level and area curves in the North Aral started to elevate after 2005. Moreover, the amount of water storage in the Aral Sea Region declined from 2002 to 2016.

The coupling degree model overall considers the variations in surface water and farmland. In addition, the average coupling degree value exhibits high-quality coupling coordination, demonstrating that farmland changes significantly impact the surface water.

Meanwhile, the volume of water loss by abandoned farmland caused additional loss of normal domestic water for people. Such a method can be referenced to investigate the effect of farmland on surface water or other fields. Moreover, the mentioned results are capable of helping relevant researchers gain insights into the effect of variations in farmland on surface water deterioration in this region, and offer sufficient support for subsequent governance and research.

### Funding
The authors received funding from the National Natural Science Foundation of China: 41971310. The funders had no role in study design, data collection and analysis, decision to publish, or preparation of the manuscript.

### Grant Disclosures
The following grant information was disclosed by the authors:
National Natural Science Foundation of China: 41971310.

### Competing Interests
The authors declare that they have no competing interests.

Jiancong Shi is employed by Taiyuan Design Research Institute Group Co., Ltd for Coal Industry.

### Author Contributions
- Jiancong Shi conceived and designed the experiments, performed the experiments, analyzed the data, prepared figures and/or tables, and approved the final draft.
- Qiaozhen Guo conceived and designed the experiments, authored or reviewed drafts of the paper, and approved the final draft.
- Shuang Zhao conceived and designed the experiments, prepared figures and/or tables, and approved the final draft.
- Yiting Su analyzed the data, authored or reviewed drafts of the paper, and approved the final draft.
- Yanqing Shi analyzed the data, prepared figures and/or tables, and approved the final draft.

### Data Availability
The raw measurements are available in the Supplemental Files.

### Supplemental Information
Supplemental information for this article can be found online at http://dx.doi.org/10.7717/peerj.12920#supplemental-information.

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
