# Peer review of "The effect of farmland on the surface water of the Aral Sea Region using Multi-source Satellite Data"

_PeerJ, doi:10.7717/peerj.12920_

## Round 0.1 · original submission · Major Revisions

Dear Dr. Shi,

Thank you for your submission to PeerJ.

Reviewers' comments on your work have now been received. The manuscript has been assessed by three reviewers. Reviews indicated that the novelty of the study, method, discussion and reference is not sufficient. I agree with this evaluation and I would, therefore, request for the manuscript to be revised accordingly.

Reviewer 3 has suggested that you cite specific references. You may add them if you believe they are especially relevant. However, I do not expect you to include these citations, and if you do not include them, this will not influence my decision.

My suggested changes and reviewer comments are shown below and on your article 'Overview' screen. In addition, one of the reviewers has attached an annotated manuscript to this review.

With kind regards,
Chenxi Li
Academic Editor, PeerJ


·

Basic reporting

1. This article conformed to professional standards of expression.
2. This article referenced relevant literatures and provided sufficient background.
3. The structure of the article was reasonable. The content of line 278 (3.3 Accuracy assessment) should be located before 3.1 and 3.2, I think. Only the accuracy assessment reached the satisfied degree, the analysis to the farmland and waters variations were dependable.

Experimental design

Multisource satellite data is being used in many fields, and the utilization in the field of water in the article is a useful attempt. The methods in the article described with sufficient details, and can provide help to the relevant research.

Validity of the findings

The multisource satellite data used in the article are robust, the data processing methods are reasonable, and then the conclusions ate crecdible.

Additional comments

1. Examining in the pater title is suggested to delete.
2. Farmland irrigation in the paper title and abstract is inaccurate. No farmland irrigation information, such as irrigation area, irrigation type, etc.,is provided in the paper. Based on the content (abandoned, replanted farmland) in the paper, land utilization maybe more accurate than farmland irrigation. And the related content in the paper can be adjusted accordingly.
3. In Line 249, water variations maybe inaccurate, and waters variations maybe more accurate.
4. Figure 7, different line types are suggested to be used to make the figure more clearly expressed.

Reviewer 2 ·

Basic reporting

no comment

Experimental design

no comment

Validity of the findings

no comment

Additional comments

See attached PDF

This paper assess the effect of irrigation on surface water. Multi-source data were employed to extract the variations of water area, water level, water storage in the Aral Sea Region. Using this data, relationship between surface water and irrigation is developed. In general, the manuscript is well explained. However, there are some issues associated with the quality of this work that need to be addressed prior to publication:

1. In line 58, you mentioned that it was verified that the reduction of downstream runoff is not associated with the upstream of the Amu Darya River and the Syr Darya River. It was verified by whom? Kindly give reference here.
2. Kindly consider rewriting sentences like in line 50 to 52 “To explore the reasons for the shrinking of the Aral Sea, numerous researches have been conducted on the rivers’ source, as presented below.”
3. In line 85 “At present, it has become an essential measure of long-term monitoring variations in surface water in large areas (Singh et al., 2012)”. What it indicate in this sentence?
4. Kindly go through your paper and remove all typos and grammar related mistakes.

Annotated reviews are not available for download in order to protect the identity of reviewers who chose to remain anonymous.

Reviewer 3 ·

Basic reporting

The paper is overall clear and cited relevant references in most places needed. The paper follows a professional article structure. However, there are places that need improvement.

1. Abstract:
a. “Therefore, the surface water of the Aral Sea continuously deteriorated each year.” How can the increase in drainage deteriorate the surface water? I do not understand how the “therefore” plays a role here. In addition, the “deteriorate” is confusing. It sounds like you are talking about water quality; however, you only consider water quantity.
b. You mentioned irrigation methods/models. This is very confusing. To me, irrigation methods are drip irrigation, sprinkling irrigation, surface irrigation, etc. Apparently, the paper does not consider different types of irrigation methods.
c. I suggest adding implications at the end of the abstract, e.g., who can benefit from your study?
2. Lines 58-59, reference is needed.
3. Line 75, what do you mean by “establish a scientific method”? Are the methods in the previous studies not scientific? The statement is too general and arbitrary.
4. Line 194, I am wondering about the uncertainties of the three datasets about water height. Why is a simple average good enough to be used here? What if one of them has better quality/ less uncertainties?
5. You calculated water storage using equation (5). Then what about the water storage from GRACE data? It is unclear which data/calculation you used to reflect water storage in later analysis.
6. Line 202, why do you only validate the 1987 classification results? What about other years?
7. Lines 219 and 222, what are the xb, xab?
8. Line 244-245, based on what evidence, did you state that the decrease in runoff is impacted by replanting of the farmland? I thought this is something you need to demonstrate in later analysis.
9. Line 254, where is the Dike Kokaral dam? Why can it reduce the water flowing into East Aral?
10. Line 287, availability should be “capability”?
11. Line 302, wtaer is a typo.
12. Figure 1, is “Reacher” a typo?

Experimental design

Some details of the methodology are unclear or not robust.

1. I do not think the sample sizes are robust for the correlation analysis (Figs. 8 and 9). There are only seven samples. Would it be possible to conduct the correlation using the values of each grid cell instead of an annual value? In addition, the values shown in Figs. 8 and 9 are based on the classification using random forest, so the correlation is afterthought analysis. Unless you have independent observations or mechanism hypotheses to verify your results, I do not see the reason why the correlation analysis can test the reliability of water extraction results.

2. Details of the random forest are unclear to me. First, how did you select the 200 training samples? Will the different choices of the samples influence the validation results? Second, what is the value for ntree? How are the validation results sensitive to parameter tuning? Third, mtry is the square root value of the number of variables. What are the variables?

Validity of the findings

I do not think the paper has good novelty.

1. The authors highlight that their contribution is establishing the real connection between irrigation and surface water, and the method they use is the entropy method. First, it is unclear what criteria they use to determine that the coupling coordination is high-quality. Second, the coupling degree values quantified in this study can only verify the importance of irrigation on surface water in the Aral Sea Region. However, such importance has been observed and reported by numerous previous studies. To name only a few:
Shibuo, Yoshihiro, Jerker Jarsjö, and Georgia Destouni. "Hydrological responses to climate change and irrigation in the Aral Sea drainage basin." Geophysical Research Letters 34.21 (2007).
Micklin, Philip. "The past, present, and future Aral Sea." Lakes & Reservoirs: Research & Management 15.3 (2010): 193-213.
Cretaux, Jean-François, René Letolle, and Muriel Bergé-Nguyen. "History of Aral Sea level variability and current scientific debates." Global and Planetary Change 110 (2013): 99-113.
I do not see the value of the coupling degree values. For example, how can the coupling degree value you quantified here provide implications to water resources management? I would say more innovative or quantitative research could be to specify the contribution of irrigation to the water fluxes, i.e., how much water decreases are due to irrigation and how such amount varies over time.

2. The discussion is very general, especially in section 4.2. The effect of farmland irrigation on surface water is the contribution of this paper, as the authors highlighted in the introduction. However, the discussion is just a repeat of what they have done. I suggest the authors discuss the influencing mechanism of irrigation on surface water in the Aral Sea. In addition, it is necessary to compare their results with previous literature to highlight their contribution or to verify the reliability of their results.

---

## Round 0.2 · Major Revisions

Reviewers' comments on your work have now been received. Reviewers indicated that the mechanisms of abandoned and replanted farmland leading to decreases of the Aral Sea were unclear. In addition, abstract, introduction, results, discussion section should also be improved. Moreover, The typesetting of papers also needs to be improved. I agree with this evaluation and I would, therefore, request for the manuscript to be revised accordingly.

Reviewer 3 ·

Basic reporting

no comment

Experimental design

no comment

Validity of the findings

no comment

Additional comments

I am overall satisfied with the authors’ responses. Many thanks to the authors for providing further analysis. I have several additional comments as follows.

Major comments:
1. The mechanisms of abandoned and replanted farmland leading to decreases of the Aral Sea are unclear to me. Abstract mentions the “Two types of farmland changes… led area of the Aral Sea to decrease”. Please clarify the mechanism separately for abandoned farmland and replanted farmland in the discussion section, i.e., how they can both lead to the decreases of the Aral Sea. To my understanding, abandoned and replanted farmland are two opposite ways of farmland utilization. It is unclear why they lead to the effects of the same direction on the Aral Sea. If they both lead to a decrease, which leads to a greater decrease, and why?
I notice that the authors try to discuss it in section 4.2. Unfortunately, this section is confusing. The authors mentioned that “if abandoned farmland were used to plant cotton, the water loss would reach 0.028 km3 every year”, and such an amount of water loss could provide water use for tens of thousands of people. To my understanding, the authors are trying to stress that farmland/cropland leads to a great amount of water loss. In other words, if the abandoned farmland were not used for planting, the water loss would not be that great. However, later in the text, the authors mentioned: “abandoned farmland has aggravated the loss of water”. This is contradictory to the previous evidence. In addition, it is unclear what the “water wasted” the authors refer to, is it the same as water loss?
2. The description of the selection of training samples is still unclear. In line 195, “the training samples were selected according to the high-resolution image of Google Earth (Figure 3).” The “according to” is confusing to me. Does it mean that “the high-resolution image of Google Earth” is a criterion you used to select training samples? If so, based on what criteria did you determine it is high-resolution? Did you randomly choose the 200 training samples from all the high-resolution images of Google Earth?

Minor comments:
Abstract:
Line 17, “unscientific” should be “improper”?
Lines 18 and 36, sentences usually do not start with “And”.
Line 36, “coupling coordination between irrigation and surface water” should be between “surface water” and “farmland”?
Line38, “The findings are contributed for …”, delete “are”. Change “formulate” to “formulating”.
Line41, What “ideas”? This is very ambiguous. Please describe directly what insights or intellectual merits your work provides to the academic community.

Introduction:
Line 133, consider deleting “to address the above issues”.

Results:
Line 287: Consider mentioning the Syr Darya and the Amu Darya in the caption of Figure 3. In addition, what do you mean by “mainly diffused especially around the Amu Darya”? Are you looking at the percentage of abandoned farmland in each Darya River? If so, suggest adding the number or percentage in the text.
Lines 289-291: The sentences are very unclear to me. First, I can’t see why Fig. 4 shows the annual change peaked between 1987 and 1993. To me, the figure shows the area of abandoned farmland, but the colors between the years are very difficult to distinguish. It may be better to indicate the values of abandoned farmland for each year in the label. Second, what do you mean by “abandoned rate”? Is it the rate of change in abandoned farmland between years (e.g., 2002 to 2016), or the percentage of abandoned farmland in a given Darya in each year from 2002 to 2016? Third, “the abandoned rate was elevated by 5.7% in 2017”. What baseline are you compared to when you say “elevated by 5.7%”?
Line 297: Again, I can’t determine the replanted rate for each year from Figure 5, let alone the trend of the rate.
Fig. 8, The figure also shows equivalent water height, but it is not described in the text.

Discussion:
Section 4.1 is like a summary of what you did instead of a real discussion. Discussion should discuss how the results can be interpreted from the perspective of previous studies. For example, do your results contradict or coincide with previous studies, and why?
Line 379, “Most e previous”, “e” is redundant.

Others:
The paper lacks space between words in many places. To name only a few, line 70, “landsto” should be “lands to”. Line 122, “Regionis” should be “Region is”. I suggest authors carefully check the spaces throughout the paper.

---

## Round 0.3 · Minor Revisions

Reviewers' comments on your work have now been received. The manuscript has been assessed by two reviewers. Reviewers indicated that the sub-headlines, tables, figures and several type writings need to be improved. I agree with this evaluation and I would, therefore, request for the manuscript to be revised accordingly.

·

Basic reporting

1. This article conformed to professional standards of expression.
2. This article referenced relevant literature and provided sufficient background.
3. The structure of the article was reasonable.

Experimental design

Multi-source satellite data is being used in many fields, and the utilization in the field of water area analysis in the article is a useful attempt. The methods in the article described with sufficient details, and can provide help to the relevant research.
The multi-source satellite data used in the article are robust, the data processing methods are reasonable, and then the conclusions ate credible.

Validity of the findings

1.In line 165,2.3.1 Detection Algorithm of Farmland Change, maybe more accurate.
2.In line 183,2.3.2 Detection Algorithm of waters (or water area) Change, maybe more accurate. Please pay attention to the difference between water and waters.
3.Line 266, Table should be located behind related text.
In Figure 3, a1,a2, bi,b2,b3, is suggested to be located in the bottom of the figures.

Reviewer 3 ·

Basic reporting

no comment

Experimental design

no comment

Validity of the findings

no comment

Additional comments

I am quite pleased with the authors' responses to my comments. I appreciate the authors’ work. The paper is appropriate to publish at PeerJ after addressing the following minor comments.

Line 294, change “by” to “to”.
Lines 331 and 332, please change “Water mass” to lowercase.
Line 362, “schoolers” should be “scholars”.
Line 425, “disturbed” should be “disturbance”.
Lines 428-429 and Line 432, “it is necessary to identify whether or not need to add other indices for improving accuracy.” and “we aim at more complete data to observe the change of the surface water”. Please fix the grammar.

---

## Round 0.4 · accepted · Accept

The authors have adequately addressed the comments raised in the previous round of review and I feel that this manuscript is now acceptable for publication.